# Alleviating Effects of Methyl Jasmonate on Pepper (*Capsicum annuum* L.) Seedlings under Low-Temperature Combined with Low-Light Stress

**DOI:** 10.3390/plants13192694

**Published:** 2024-09-26

**Authors:** Kaiguo Pu, Nenghui Li, Yanqiang Gao, Miao Zhang, Tiantian Wang, Jianming Xie, Jing Li

**Affiliations:** College of Horticulture, Gansu Agricultural University, Yingmen Village, Anning District, Lanzhou 730070, China; pukaiguo1998@163.com (K.P.); linh@st.gsau.edu.cn (N.L.); zlm20000712@163.com (Y.G.); zm15609379816@163.com (M.Z.); 17789365893@163.com (T.W.)

**Keywords:** pepper, low temperature combined with low light, MeJA, photosynthetic properties, antioxidant

## Abstract

Low temperature combined with low light (LL) is an important factor limiting pepper quality and yield. ‘Hang Jiao No. 2′ were used as experimental materials, and different concentrations of MeJA (T1 (0 μM), T2 (100 μM), T3 (150 μM), T4 (200 μM), T5 (250 μM) and T6 (300 μM)) were sprayed under LL stress to explore the positive effect of exogenous methyl jasmonate (MeJA) on peppers under LL stress. The photosynthetic properties, osmoregulatory substance, reactive oxygen species, antioxidant enzyme activities, and related gene expressions of the peppers were measured. Our results demonstrated that 200 μM MeJA treatment significantly increased chlorophyll content, light quantum flux per active RC electron transfer (Eto/RC), maximum captured photonic flux per active RC (TRo/RC), energy flux for electron transfer in the excitation cross section (Eto/CSm), energy flux captured by absorption in the excitation cross section (TRo/CSm), soluble protein, and soluble sugar content. Moreover, it significantly improved the maximum photochemical efficiency of PSII (Fv/Fm) and performance index based on absorbed light energy (PI (abs)) by 56.77% and 67.00%, respectively, and significantly decreased malondialdehyde (MDA) content and relative conductivity by 30.55% and 28.17%, respectively. Additionally, antioxidant enzyme activities were elevated, and the expression of the related genes was activated in pepper seedlings under stress, leading to a significant reduction in reactive oxygen species content. In conclusion, our findings confirmed that 200 μM MeJA could reduce the injury of LL to pepper leaves to the photosynthetic organs of pepper leaves, protect the integrity of the cell membrane, and further improve the tolerance of pepper seedlings to LL.

## 1. Introduction

Abiotic stresses are primarily attributed to environmental factors such as high salt concentrations, extreme temperatures, low light, and drought [1], inhibiting normal physiological functions, adversely affecting growth and metabolism [2], and even injuring cells, leading to plant death and low yields [3]. When plants are exposed to stress, they produce large quantities of reactive oxygen species (ROS) and free radicals. Excess accumulation of these molecules can have toxic effects on cells. The plant body activates active precautionary systems to ensure growth and development.

Pepper (*Capsicum annuum* L.) is a thermophilic and heliophilic vegetable native to South America and one of the main vegetables grown in facilities. Low temperature combined with low light (LL) is a significant factor restricting the yield of peppers grown in facilities in Northwest China during winter and spring [4]. LL results in the inhibition of the expression of genes related to chlorophyll metabolism and the post-translational modification of enzymes, ultimately leading to decreases in chlorophyll content and net photosynthetic rate [5,6]. However, LL significantly increases the content of malondialdehyde (MDA), alters osmoregulatory substances, and increases cell membrane permeability [7], all of which deteriorate the normal physiological functions of peppers. Antioxidants are important in maintaining cellular redox homeostasis and low-temperature tolerance [8]. LL stress causes oxidative damage to plants, activates antioxidant mechanisms, and enhances the antioxidant capacity of peppers to restore their normal physiological functions and improve their stress resistance. In addition, various metabolic and physiological activities of plants are linked to phytohormones, which play a crucial role in triggering complex processes such as plant growth, development, and stress responses by responding to signaling cascades in plants [9]. Therefore, improving the LL tolerance of peppers in facilities has become key to the early spring cultivation of peppers and improving yields.

Phytohormones are the main mediators of plant tolerance to abiotic stress under adverse climatic conditions [10,11]. Jasmonic acid (JA) is a plant signaling molecule that plays an important role under both stress and normal conditions. Under stress, it is mainly associated with activating the antioxidant system, stomatal movement, and amino acid and sugar synthesis in plants [12]. Methyl jasmonate (MeJA), an ester of JA, has a wide range of physiological effects on plant growth [13] and is closely related to plant resistance, acting as an endogenous signaling molecule involved in stress responses such as mechanical injury, salt stress, and low temperature. Studies have shown that exogenous spraying of MeJA effectively improves plant tolerance to low-temperature stress [14]. Existing studies on MeJA have focused on the role of MeJA in anti-stress responses, such as insect and disease resistance, whereas little has been reported on the effect of MeJA on pepper hardiness. Therefore, in this study, pepper seedlings pretreated with exogenous MeJA were used as test materials, and LL stress was simulated using an artificial constant-temperature climate chamber to study the response mechanism of exogenous MeJA on the LL of pepper seedlings and to provide a theoretical basis for pepper cultivation and management in winter and spring seasons.

## 2. Results

### 2.1. Effect of Exogenous MeJA on Growth Indexes of Pepper Seedlings under LL Stress

There was a significant increase of 4.50%, 5.25%, and 4.50% in plant height in the T2, T4, and T5 treatments compared with that of T1, respectively, after 7 d of stress treatment. Conversely, no significant differences were observed between treatments T3 and T6. There was no significant difference in stem thickness for any treatment compared to that of T1 (Figure 1A).

Under LL stress, the dry matter content of pepper seedlings tended to increase and then decrease with increasing exogenous MeJA concentrations, with the T4 treatment having the highest dry matter mass. The aboveground fresh weight and belowground fresh weight of T4, T5, and T6 were significantly increased by 44.05%, 20.69%, and 19.55% and 53.27%, 34.36%, and 34.28%, respectively, when compared to that of T1 (Figure 1B). The aboveground dry weight and belowground dry weight of T4 were significantly higher than those of T1 at 26.17% and 48.45%, respectively, while the rest of the treatments did not show significant differences (Figure 1C).

### 2.2. Effects of Exogenous MeJA on Malondialdehyde (MDA) and Relative Conductivity of Pepper Seedlings under LL Stress

As the concentration of exogenous MeJA increased, both the relative conductivity and MDA content first decreased and then increased (Figure 2). The MDA contents of the T3, T4, and T5 treatments were significantly lower (16.07%, 30.55%, and 21.03%, respectively) than that of the T1 treatment (Figure 2A), and the relative conductivities of all treatments were significantly lower (10.18%, 26.98%, 28.17%, 10.30%, and 6.18% for T2 to T6, respectively) than that of T1 with an increase in MeJA concentration (Figure 2B).

### 2.3. Effects of Exogenous MeJA on the Superoxide Anion Content of Pepper Seedlings under LL Stress

Qualitative analysis of superoxide anion content in pepper leaves was conducted us-ing nitro tetrazolium blue chloride (NBT) histochemical staining. As the concentration of MeJA increased, the staining became lighter and then darker, with the T4 treatment showing the lightest staining (Figure 3A). The O_2_^−^ content showed a trend of decreasing and then increasing with an increase in the MeJA concentration. The O_2_^−^ content of treatments T3, T4, and T5 decreased significantly (30.29%, 46.34%, and 20.23%, respectively) compared to that of T1, and the T4 treatment O_2_^−^ content was the lowest (Figure 3B).

### 2.4. Effects of Exogenous MeJA on Photosynthetic Capacity of Pepper Seedlings under LL Stress

Under a combination of low temperature and low-light stress, the Chl a, Chl b, and Chl a+b content of pepper leaves first increased and then decreased with an increase in exogenous MeJA concentration (Figure 4A–C). Chl a content was significantly higher in treatment T2, T3, and T4 (7.92%, 15.26%, and 19.54%, respectively) than in T1 treatments, while it was significantly lower (8.46%) in T6 (Figure 4A). Chl b content was significantly higher in treatments T2, T3, and T4 (6.23%, 13.40%, and 16.05%, respectively) than in T1, while it was significantly lower (9.26%) in T6 (Figure 4B). Chl a+b content was significantly higher in treatments T2, T3, and T4 (7.51%, 14.81%, and 18.69%, respectively) than in T1 (Figure 4C).

As shown in Table 1, Fv/Fm, Y(II), and qP were significantly higher in the T3, T4, and T5 treatments than in the T1 treatment; Fv/Fm was elevated by 21.48%, 56.77%, and 26.11%, Y(II) was increased by 19.23%, 25.96%, and 9.42%, and qP increased by 36.58%, 90.06%, and 43.14%, respectively. Meanwhile, Y(II) significantly increased by 10.38% in T2 compared with that in T1. Compared with the T1 treatment, NPQ significantly decreased by 28.57%, 73.10%, and 43.34%; qN significantly decreased by 13.75%, 32.60%, and 23.91%; and 1-qP significantly decreased by 35.57%, 87.57%, and 41.95% under treatments T3, T4, and T5, respectively. T6 treatment showed a significant decrease of 28.19% in NPQ, a significant increase of 5.89% in qN, and a significant decrease of 16.24% in 1-qP compared to that in T1. The excess excitation energy ((1-qP)/NPQ) of pepper leaves significantly decreased by 52.76% in T4 compared to that in T1. Furthermore, the chlorophyll fluorescence parameters were consistent with the results of the aforementioned visual analysis (Figure 5).

Chlorophyll fluorescence kinetic curves were determined for each treatment to understand how MeJA alleviates the photoinhibition of PSII induced by low temperature combined with low-light stress (Figure 6A). The results showed that the J-I-P stage of all treatments with exogenously sprayed MeJA was higher than that of the low-temperature combined with low-light stress treatments, of which the J-I-P stage was the highest in T4. Technical fluorescence parameters were obtained by JIP-test analysis of measured OJIP curves (Figure 6B). Compared with the T1 treatment, φ0, Sm/t(Fm), φP0, φE0, φR0, and PI abs showed a tendency to increase and then decrease with the increase in sprayed MeJA concentration in each stress treatment. All parameters were most significant in the T4 treatment, with significant increases of 0.61, 0.27, 0.30, 0.43, 1.06, and 0.67 times. In contrast, M0, Vj, Vi, Sm, and δR0 showed a decreasing and then increasing trend, all of which were most significant in T4 (16.14%, 14.17%, 27.05%, and 25.49%, respectively).

Functional parameters obtained from the JIP analysis were used to compare the changes in the energy allocation of individual active reaction centers in pepper leaves treated with different MeJA concentrations to further understand the photosynthetic behavior of pepper leaves under different concentrations of MeJA (Figure 6C,D). ABS/RC and DIo/RC in all treatments with MeJA application significantly decreased and then increased, with the most obvious decreases in the T4 treatment, which decreased by 18.44% and 27.35%, respectively, compared with that of T1. In contrast, both TRO/RC and ETo/RC showed an increasing and then decreasing trend in all treatments compared with the T1 treatment, with the most significant increase in the T4 treatment (Figure 6C). ABS/CSm, TRo/CSm, and ETo/CSm all increased and then decreased under different concentrations of MeJA treatment, and all peaked concentrations in T4, which increased by 14.91%, 27.15%, and 56.84%, respectively, compared to that of T1. In contrast, DIo/CSm showed a decreasing and then increasing trend; the T4 treatment was the smallest, with a significant decrease of 9.53% compared to that of the T1 treatment (Figure 6D).

### 2.5. Effects of Exogenous MeJA on Soluble Protein and Soluble Sugars in Pepper Seedlings under LL Stress

Under combined low temperature and low-light stress, both soluble protein and soluble sugar contents tended to increase and then decrease with increasing MeJA concentration, and both were highest in the T4 treatment. The soluble protein content of all treatments was significantly higher than that of T1 (25.17%, 34.97%, 68.48%, 41.16%, and 46.27% for T2 to T6, respectively) (Figure 7A). The soluble sugar content was significantly higher than that of T1 (12.34%, 34.80%, 65.03%, 25.62%, and 9.22% for T2 to T6, respectively) (Figure 7B).

### 2.6. Effects of Exogenous MeJA on Antioxidant Enzyme Activities and Related Gene Expressions in Pepper Seedlings under LL Stress

Under combined low temperature and low-light stress, superoxide dismutase (SOD), peroxidase (POD), and catalase (CAT) enzyme activities showed a trend of increasing and then decreasing with increasing MeJA concentration (Figure 8A–C). SOD enzyme activity was significantly higher in all treatments than in T1 (15.50%, 28.01%, 55.69%, 34.49%, and 29.52% in T2 to T6, respectively) (Figure 8A). POD enzyme activity increased by 11.58%, 14.12%, 19.90%, 16.32%, and 6.47% in T2 to T6, respectively, compared to that in the T1 treatment (Figure 8B). Compared to that in T1, CAT enzyme activity was significantly increased by 13.91%, 23.68%, and 46.98% in T2, T3, and T4 treatments, respectively, while in T6 treatment, CAT enzyme activity was significantly decreased (12.47%) (Figure 8C). *CaSOD*, *CaPOD*, and *CaCAT* genes were all up-regulated and then down-regulated with an increase in MeJA concentration (Figure 8D–F). *CaSOD*, *CaPOD*, and *CaCAT* gene expressions were significantly up-regulated 1.69 times, 2.18 times, and 80.20%, respectively, in the T4 treatment compared to that in the T1 treatment; *CaSOD*, *CaPOD,* and *CaCAT* gene expression in T6 treatment have no significant difference with T1 treatment.

## 3. Discussion

Low temperature combined with low light (LL) represents a complex form of abiotic stress that can significantly impede the physiological and biochemical processes of plants, thereby causing a phenomenon known as ‘photoinhibition’ and oxidative damage, which has a deleterious effect on plant growth and developmental processes, as well as the potential yields [15]. Methyl jasmonate MeJA enhances plant resilience to abiotic stressors, prompting a cascade of physiological and metabolic alterations that diminish oxidative damage by augmenting antioxidant enzyme activity [16]. The morphological changes observed in plants subjected to stress directly reflect their tolerance to stress [17]. The results of this study indicated that pepper seedlings sprayed with MeJA exhibited significantly elevated dry matter mass compared to those in the untreated control group under LL conditions. This result aligns with the findings of Li et al. [18], who reported that the aboveground dry weight of pepper seedlings treated with exogenous oleuropein lactones was significantly greater than that of the control treatment after exposure to low-temperature stress.

The relative conductivity and malondialdehyde (MDA) content of the leaves serve as important indicators of the extent of damage to leaf cell membranes [19]. In response to low-temperature stress, damage to plant leaf cell membranes results in the efflux of electrolytes, which in turn increases conductivity and MDA content [20]. In a study conducted by Li et al., the application of exogenous melatonin mitigated the inhibitory effects of low temperature and light stress on the growth of pepper seedlings by reducing damage to cell membranes [4]. Tang et al. [16] studied the effects of exogenous zeaxanthin on pepper seedlings under low-temperature stress and obtained similar results. This is consistent with the results of the present study, in which the relative conductivity and MDA content of pepper seedling leaves were significantly reduced by MeJA pre-treatment under stress and the most pronounced in the 200 μM MeJA treatment. This may be due to the ability of MeJA to mitigate the membrane lipid peroxidation induced by the combination of LL and preserve the integrity of plant cell membranes [21].

Chlorophyll is responsible for the absorption, transfer, and conversion of light during photosynthesis. The chlorophyll concentration in plants directly affects photosynthesis and is one of the most important physiological indicators of plant tolerance to abiotic stress [22]. In this study, exogenous spraying of MeJA significantly increased the Chl a, Chl b, and Chl a + b contents of pepper seedlings under LL stress. Similar results were found by Yu et al. [23], who found that exogenous spraying of melatonin enhanced LL tolerance in eggplant seedlings by increasing the total chlorophyll content and net photosynthetic rate compared to the control. MeJA may increase the activity of chlorophyll synthase under low temperatures combined with low light, or it may inhibit stress-induced chlorophyll degradation, leading to an increase in chlorophyll content.

Chlorophyll fluorescence has been widely used as a non-destructive and effective method for assessing the effects of abiotic stresses on photosynthetic electron transport systems [24]. Zhu et al. [25] found that low temperature and light reduced the ability of pepper seedling leaves to capture and use light energy and caused changes in light energy allocation. In this experiment, exogenous MeJA pretreatment of the Photosystem II (PSII) reaction center resulted in an increase in quantum yield for photosynthetic electron transfer, a decrease in quantum yield via heat dissipation, and a decrease in non-regulated quantum yield, which is in agreement with the findings of Yu et al. [26] that more tolerant plants had higher photosynthetic quantum yield. This suggested that MeJA can regulate the quantum yield of PSII in pepper seedlings under low temperatures combined with low light and alleviate photoinhibition. In addition, the maximum photochemical efficiency of PSII (Fv/Fm) reflects the photoinhibition status of plant PSII and can be an important indicator of plant adversity tolerance, which is reduced by stress [27]. Rácz et al. [28] revealed that pepper varieties tolerant under stress had higher photochemical burst factor (qP) and lower non-photochemical burst (NPQ). Consistent with the above studies, qP was elevated, and NPQ was reduced in MeJA pretreated pepper leaves before low temperature combined with low-light stress in the present study, with the 200 μM MeJA treatment being the most significant. Moustakas et al. [29] showed that melatonin attenuates low-temperature-induced PSII photoinhibition by reducing excess excitation of photosystems, which is consistent with the results of the present study; PSII excitation voltage (1-qP) and surplus stimulus energy ([(1-qP)/NPQ]) decreased in pepper seedlings under low temperature combined with low-light stress after MeJA pretreatment, whereas Fv/Fm increased significantly.

The fast chlorophyll fluorescence kinetic curve reflects changes in the PSII reaction center primary photochemistry, state of the photosynthetic machinery, and other changes mainly on the donor side, acceptor side, and reaction center of PS II [30]. The results showed that MeJA alleviated the decrease in the OJIP curve of pepper under low temperature combined with low-light stress, indicating that MeJA alleviated the inhibition of electron transfer from Q_A_ to Q_B_ and exerted a protective effect on the PSII receptor. The O-J, J-I, and I-P stages were lower than those in the MeJA treatment under low temperature combined with low-light stress. This may be due to the low temperature combined with low-light stress, reducing the efficiency of the fast reducibility of the PQ pool during electron transfer, leading to a decrease in the I-phase, as well as the stress-disrupting chlorophyll proteins on the side of the PSI receptor [31]. However, the results of the present study showed that the J phase was not obvious in all treatments, which might be caused by the accelerated reduction rate of the fast reducibility PQ pool in the pre-J-I phase or the accelerated reduction rate of the PSII primary electron acceptor Q_A_ in the O-J phase of the photochemical reaction due to the low temperature combined with low-light stress. In addition, pepper plants treated with MeJA attenuated the extent of inhibition of J, I, and P phases by LL stress. We speculated that MeJA may have contributed to inhibiting the Q_A_ pool shrinkage or blocked the ability to transfer electrons from the donor side of PSII. The JIP test can be used to identify parameters such as energy uptake, trapping, and electron transfer in PSII and PSI, and it is widely used in plant responses to stress conditions [32]. The quantum yield of electron transport flux from Q_A_ to Q_B_ (φEo) and quantum yield of PSII final electron acceptor reduction per photon absorbed (φRo) decreased under salt stress, suggesting that salt stress disrupts the damage of primary photochemical reactions in tomato leaves, inhibiting electron transfer on the PSII receptor side [33]. Similar to the present experiment, the values of φEo, maximum quantum yield of primary PSII photochemistry (φPo), φRo, and efficiency of electron transfer from Q_B_ to PSI receptors (δRo) were significantly higher in the MeJA treatment than in the stress treatment in the present experiment, and were most significant for 200 μmol·L^−1^ MeJA, indicating that the MeJA treatment increased the photochemical efficiency, electron transport flux, and the quantum efficiency of PSII to PSI in PSII under low temperature combined with low-light stress. PI (abs) is an important indicator for evaluating plant health under adversity [34]. PSII, one of the most sensitive components of photosynthesis, causes a significant reduction in the electron transport chain under abiotic stress [35]. The results of this experiment showed that PI (abs) decreased, average absorbed light quantum flux per PSII reaction center (ABS/RC) and light quantum flux dissipated per active RC (DIo/RC) increased significantly, and maximum captured photonic flux per active RC (TRo/RC) and light quantum flux per active RC electron transfer (ETo/RC) decreased significantly in pepper seedlings under low-temperature stress, which was attributed to the fact that stress led to a large number of RC being in an inactivated state, which reduced the efficiency of PSII energy capture and electron transfer [36], and this state of affairs was broken by MeJA. TRo/CSm and ETo/CSm decreased under stress and inhibited light energy capture by the PSII reaction center [37]; similar to in the present study, the significant increase in DIo/RC and ABS/RC values and the significant decrease in TRo/RC and ETo/RC values in the LL treatment might be attributed to the fact that LL stress caused most of the RCs to be in an inactivated state, which, in turn, reduced the energy capture efficiency and electron transport efficiency of PSII [36]. MeJA pretreatment significantly increased TRo/RC, energy flux captured by absorption in the excitation cross section (TRo/CSm), ETo/RC, and energy flux for electron transfer in the excitation cross section (ETo/CSm) and significantly decreased DIo/RC and energy flux dissipated in the excitation cross section (DIo/CSm), which resulted in improved energy uptake efficiency of PSII, enhanced light energy capture and electron transfer in the PSII reaction center, and improved low-temperature combined with low-light tolerance of pepper.

Chloroplasts are the primary sites of reactive oxygen species (ROS) production in plants [38]. Oxidative stress occurs when ROS production exceeds degradation [39]. Abiotic stresses, such as LL, not only inhibit plant growth and development but also cause excessive accumulation of ROS, leading to oxidative damage and even the death of plant tissues [40]. In the present study, the O_2_^−^ content of pepper leaves decreased and then increased with increasing MeJA concentrations. O_2_^−^ quantification and NBT histochemical staining of pepper leaves showed that MeJA was important in inhibiting reactive oxygen species (ROS) accumulation under LL stress, suggesting that MeJA can protect plants from stress by controlling O_2_^−^ and thiobarbituric acid reactants (TBARS) levels [41]; antioxidant enzymes and non-enzymatic antioxidants are the major ROS scavenging forces [42,43,44]. Wang et al. [45] found that the protective enzyme activities were lower in cold-tolerant microphytobenthos, but the protective enzyme activities and antioxidant contents were also higher in cold-tolerant plants, concluding that antioxidant enzyme activities are positively correlated with cold tolerance. Tang et al. [16] showed that zeaxanthin pretreatment significantly increased SOD, POD, and CAT activities of peppers under stress. Li et al. [18] also showed that exogenous oleuropein lactones attenuated oxidative damage induced by low-temperature stress by increasing the activity of antioxidant enzymes in peppers. Consistent with this study, the decrease in antioxidant enzyme activities after LL stress may be due to the disruption of the ROS homeostatic balance [46], which leads to the accumulation of H_2_O_2_ and O_2_^−^, thus exacerbating the cellular damage caused by oxidative stress. In this study, MeJA pretreated pepper seedlings increased the activities of SOD, POD, and CAT; up-regulated the expression of genes *CaSOD*, *CaPOD*, and *CaCAT*; and decreased the reactive oxygen species (ROS) level. It was speculated that MeJA promote the initiation of antioxidant mechanisms under LL stress, the increase in antioxidant enzyme activities, which in turn scavenged oxygen radicals and counteracted the deleterious effects caused by the accumulation of ROS [47]. This suggested that MeJA is involved in enhancing ROS detoxification, reducing cell damage and death, and alleviating LL induced inhibition; similar effects have been observed in plants under different stresses [48]. It is conjectured that the action of MeJA on antioxidant enzyme activity may be one of the mechanisms of oxidative stress tolerance in pepper under LL stress, and is related to the activation of pepper precautionary mechanisms under LL stress.

MeJA enhanced the antioxidant system of pepper seedlings and increased their antioxidant content. Soluble sugars and soluble proteins not only have similar effects as non-enzymatic antioxidants but are also important osmoregulatory substances [49]. The content of osmoregulatory substances is closely related to oxidative damage in plants, and LL causes oxidative damage in pepper seedlings; thus, increasing the accumulation of osmoregulatory substances is a prerequisite for improving the tolerance of pepper to LL stress. The results of this experiment showed that exogenous MeJA pretreatment promoted the accumulation of soluble proteins and sugars and reduced oxidative damage in pepper seedlings under LL stress. Therefore, it was conjectured that MeJA was involved in maintaining cellular homeostasis in pepper seedlings after stress as a result of a decrease in cellular osmotic potential and settled the protoplasmic colloid of cells, thus reducing the oxidative damage caused by the stress. Additionally, various metabolic reactions and regulations of plants directly link soluble sugars with the production rates of reactive oxygen species [50]. Thus, it was accounted that MeJA was involved in regulating ROS-producing pathways, such as mitochondrial respiration or photosynthesis regulation, that decrease ROS production. Alternatively, MeJA was involved in the interaction between soluble proteins and antioxidant enzymes and participates in the antioxidant process by regulating its activity, thus regulating the tolerance mechanism of pepper seedlings under low temperature combined with low light, decreasing the oxidative damage caused by stress to pepper seedlings, and improving the tolerance of pepper, and this hypothesis remains to be further tested.

## 4. Materials and Methods

### 4.1. Experimental Materials and Growth Conditions

The pepper seed ‘Hang jiao 2’, purchased from Shenzhou Lvpeng Agricultural Technology Company, Tianshui, Gansu, China, was selected as the test material. First, good quality, plump seeds were selected and soaked in warm water at 55 °C for 30 min, during which time the seeds were constantly stirred with a glass rod. Subsequently, they were soaked in tap water at 25 °C for 6 h and then placed on a clean towel to germinate using an artificial climatic chamber (RDN-400E-4, Ningbo, Zhejiang, China) (temperature: 28 °C, humidity: 80%, light: 0 μmol·m^−2^·s^−1^). Seeds were watered daily during the germination period, and seeds with consistent germination (1 mm) were selected and sown in plastic nutrient pots (9 × 9 cm) containing a substrate (grass charcoal/vermiculite/perlite = 3:1:1); two seeds were sown in each pot. Seedlings were cultured in an artificial climate chamber at 28/18 °C and 300 μmol·m^−2^·s^−1^ light.

### 4.2. Treatments

#### 4.2.1. Different Methyl Jasmonate (MeJA) Concentration Treatments

The experiment was conducted in a randomized block design with six treatments (T1: 0 μM MeJA; T2: 100 μM MeJA; T3: 150 μM MeJA; T4: 200 μM MeJA; T5:250 μM MeJA; T6:300 μM MeJA). Each treatment was repeated thrice with 20 plants per replicate. When the sixth leaf fully expanded, 120 mL of different concentrations of MeJA was sprayed on the whole plant.

#### 4.2.2. Alleviating Effects of Exogenous Methyl Jasmonate (MeJA) on Pepper Plants under LL Stress

Based on the above experiments, the following six treatments were set when the fifth true leaf of the seedlings was expanded:

T1 (LL, 10/5 °C, 100 μmol·m^−2^·s^−1^, 0 μM MeJA);

T2 (LL, 10/5 °C, 100 μmol·m^−2^·s^−1^, 100 μM MeJA);

T3 (LL, 10/5 °C, 100 μmol·m^−2^·s^−1^, 150 μM MeJA);

T4 (LL, 10/5 °C, 100 μmol·m^−2^·s^−1^, 200 μM MeJA);

T5 (LL, 10/5 °C, 100 μmol·m^−2^·s^−1^, 250 μM MeJA);

T6 (LL, 10/5 °C, 100μmol·m^−2^·s^−1^, 300 μM MeJA).

Foliar spraying was applied once at 20:00 every night, and the treatment was carried out continuously for 4 d followed by low temperature combined with low light treatment (temperature: 10/5 °C, light: 100 μmol·m^−2^·s^−1^). After 7 d under low temperature combined with low-light stress treatment, the functional leaves were randomly selected, rapidly frozen in liquid nitrogen, and then deposited in a −80 °C refrigerator to determine malondialdehyde, superoxide anion, soluble protein, soluble sugar, and antioxidant enzymes. The experiment was repeated three times for each treatment.

### 4.3. Determinations

#### 4.3.1. Measurements of Physiological and Biochemical Indicators

Five whole pepper plants were randomly collected and separated into aboveground and belowground, and the fresh weight was weighed. Subsequently, they were kept at 105 °C for 30 min and dried at 80 °C for 5 d, and the dry mass was weighed.

#### 4.3.2. Determination of Relative Conductivity and MDA Content of Leaves

Relative electrical conductivity (REC): Five pepper seedlings were randomly selected, and functional leaves were taken, cleaned with ultrapure water, dried, perforated with a 0.5 cm diameter hole punch, and 12 round pepper leaves were placed in clean test tubes. Then, 10 mL of ultrapure water was added and placed in a vacuum desiccator for 30 min, oscillated for 3 h (HY-8 multifunctional oscillator), and the initial conductance (S1) was measured after oscillation. The final conductance (S2) was determined after 30 min in a boiling water bath and then cooled to room temperature. Relative electrical conductivity (REC, %) = (S1 − blank)/(S2 − blank) × 100, using the conductivity value of ultrapure water as a blank [51].

MDA was determined using the 2-thiobarbituric acid (TBA) method [52]. A 10% trichloroacetic acid (TCA) extract and a mixture of 0.6% TBA with 10% TCA was prepared, and 0.5 g of frozen pepper leaf sample was weighed. Ten percent of TCA was added and extracted via centrifugation at 4000 rpm for 10 min. Subsequently, 2 mL of the supernatant was added to the mixed reaction solution, incubated in a water bath at 100 °C for 20 min, cooled, and centrifuged. The absorbance values were measured at 450, 532, and 600 nm.

#### 4.3.3. Determination of Reactive Oxygen Species Content and Histochemical Staining

Superoxide anion (O_2_^−^) content was determined using a kit from Suzhou Grace Biotechnology, and specific steps were carried out according to the manufacturer’s instructions.

O_2_^−^ was determined using histochemical staining with nitrotetrazolium blue chloride (NBT) [53]. Three leaves of pepper seedlings from different treatments were collected and placed in 150 mL wide-mouth triangular flasks. Subsequently, 50 mL of NBT staining solution (0.5 mg·mL^−1^, dissolved in phosphate buffers (PBS) at pH 7.8) was added, and vacuum treatment was performed for 30 min prior to incubation at 25 °C for 6 h. Finally, the dye solution was removed, and the decolorizing solution (lactic acid/glycerol/anhydrous ethanol = 1:1:3; *v*/*v*/*v*) was added. The samples were incubated in a boiling water bath for 5–6 min until the green color in the leaves faded completely. The stained leaves were scanned using an EPSON A3 Transparency Unit (MODEL:EU-88, Seiko Epson Corp., Tokyo, Japan) and photographed.

#### 4.3.4. Determination of Chlorophyll Content

The chlorophyll content was determined using the 80% acetone leaching method. Five functional leaves of peppers treated for 7 d were selected, punched (0.5 cm) to avoid the main leaf veins, weighed (0.1 g), added to 10 mL of 80% acetone, and macerated for 48 h. During this period, the tubes were shaken every 12 h to ensure adequate chlorophyll extraction. Finally, a spectrophotometer (UV-1780, Shimadzu Instruments (Suzhou) Co., Ltd., Suzhou, China) was used to determine the absorbance values (OD) of the extracts at 645 and 663 nm, and the chlorophyll content was calculated using the following equation:Chl a=12.71 × OD663 − 2.59 × OD645
Chl b=22.88 × OD645 − 4.67 × OD663
Chl a+b=20.29 × OD645+8.04 × OD663

#### 4.3.5. Determination of Chlorophyll Fluorescence Parameters

The chlorophyll fluorescence parameters were determined using a modulated chlorophyll fluorescence imager (IMAGIN-PAM) (Walz, Effeltrich, Germany) [54]. Three plants per treatment were randomly selected as replicates, and after 30 min of dark adaptation, the third fully expanded functional leaf was cut and secured to the assay stage using a thin wire. The detection parameters of the fluorescence imager were set as measurement light intensity of 0.1 μmol·m^−2^·s^−1^, photochemical light intensity of 81 μmol·m^−2^·s^−1^, saturated pulsed light intensity of 2700 μmol·m^−2^·s^−1^, and pulsed light time of 0.8 s, saturated pulsed light every 20 s for a total of 15 pulses. The values of the fluorescence parameters under light adaptation measured using the last saturated pulse of light were read and analyzed.

#### 4.3.6. Rapid Chlorophyll Fluorescence-Induced Kinetic Curve (OJIP) Assay

After 7 d of low temperature and light, leaf OJIP curves were determined using the plant efficiency analyzer (Handy PEA Hansatech, King’s Lynn, UK). Leaves were measured after 30 min of sufficient dark adaptation, and OJIP curves were induced by 3000 μmol·m^−2^·s^−1^. The measurement time was 1 s. The OJIP curve was analyzed using the JIP test with reference to the method of Stiebet [55], and the significance of the relevant parameters is listed in Table 2.

#### 4.3.7. Determination of Soluble Protein and Soluble Sugar

Soluble proteins were quantified using the Coomassie brilliant blue method [56]. Fresh samples were weighed (0.5 g), extracted by grinding with 5 mL of distilled water, and then centrifuged at 10,000 rpm for 10 min. A total of 1 mL of the supernatant was added to 5 mL of Coomassie brilliant blue G-250 solution, mixed thoroughly, and incubated for 2 min, and the absorbance at 595 nm was compared.

The soluble sugar content was determined using the anthocyanin colorimetry method [52]. Fresh samples (0.5 g) were weighed, extracted by grinding in 10 mL of distilled water in a boiling water bath for 10 min and filtered, and the volume was adjusted to 50 mL. A total of 0.5 mL of filtrate was added to 1.5 mL of water and 0.5 mL of Anthone chemistry. Then, 5 mL of concentrated oxalate acid was added, the samples were incubated in a boiling water bath for 1 min, cooled, and the absorbance value was recorded at 630 nm.

#### 4.3.8. Determination of Antioxidant Enzyme Activity

The 0.05 M phosphate buffer (pH 7.8) was prepared using 5 mM EDTA, 2 mM AsA, and 2% PVP. The fresh sample (0.5 g) was homogenized in 5 mL of extraction buffer and centrifuged for 20 min at 12,000 rpm at 4 °C to obtain the assay solution (enzyme solution) [49].

SOD activity: The reaction system was 3 mL, including enzyme solution (20 μL), phosphate buffer (pH 7.8, 0.05 M, 1.5 mL), methionine (130 mM, 0.3 mL), nitro tetrazolium blue chloride (0.75 mM, 0.3 mL), ethylenediaminetetraacetic acid (1 mM, 0.3 mL), riboflavin (20 μM, 0.3 mL), and distilled water (0.28 mL), and the absorbance values at 560 nm were determined using illumination at 4000 lx for 20 min.

POD activity: The reaction system was 3 mL, including enzyme solution (100 μL), guaiacol (0.3%, 2.6 mL, 0.05 M pH 6.5 phosphate buffer (PBS) dissolved), H_2_O_2_ (0.6%, 0.3 mL), and the change in absorbance at 470 nm over 2 min was recorded.

CAT activity: The reaction system was 3 mL, including enzyme solution (200 μL), H_2_O_2_ (0.067 M, 2.8 mL, 0.05 M pH 7.0 phosphate buffer (PBS) dissolved), and the change in absorbance at 240 nm over 1 min was recorded.

#### 4.3.9. Measurement of Gene Expression

Three young leaves were randomly selected from each treatment, snap-frozen in liquid nitrogen, and stored in a refrigerator at −80 °C for total RNA extraction. Total RNA was extracted from the leaves of the pepper seedlings using an RNA extraction kit (Tian Gen Biotechnology Co., Ltd., Beijing, China). The extracted RNA samples were reverse-transcribed using a cDNA kit to obtain cDNA. Gene sequences were queried and obtained from the NCBI gene website (https://www.ncbi.nlm.nih.gov/gene), accessed on 28 September 2023, and sequence comparisons were performed using NCBI BLAST. Primers were designed using NCBI Primer-BLAST (https://www.ncbi.nlm.nih.gov/tools/primer-blast), accessed on 28 September 2023, and were synthesized by Shanghai Sheng Gong Biotechnology Co., Shanghai, China. The gene sequences are listed in Table 3. The SYBR Green Priemix Pro Taq HS qPCR kit (Acres Biotechnology Ltd., Hengyang, Hunan, China) was used according to the manufacturer’s instructions. Real-time PCR was performed using a Light Cycler 96 real-time PCR system (Roche, Basel, Switzerland). The PCR cycling parameters were pre-denaturation at 95 °C for 15 min, followed by 40 cycles of denaturation at 95 °C for 10 s, and annealing/extension at 60–66 °C for 20–32 s. Each sample was measured in triplicate, actin was used as an internal reference gene, and the relative expression was calculated according to the 2^−ΔΔCt^ method.

### 4.4. Statistical Analysis

The statistical analysis of the experimental data was performed using SPSS 22.0 software, and multiple comparisons were performed using Duncan’s method (*p* < 0.05); graphics were created using Microsoft Excel 2019 and Origin 2021 Pro.

## 5. Conclusions

Exogenous MeJA treatment reduced low-temperature and light stress in pepper seedlings. MeJA at 200 μmol·L^−1^ promoted seedling growth and increased plant dry matter accumulation, maintaining cell membrane stability and osmotic balance under stress, improving photosynthetic properties and antioxidant capacity of plants, reducing oxidative damage and thus enhancing plant resilience, and effectively alleviating the damage caused by low temperature combined with low light to pepper seedlings. Therefore, applying an appropriate amount of MeJA is essential to improve the resistance of pepper and overwintering cultivation in facility peppers.

## Figures and Tables

**Figure 1 plants-13-02694-f001:**
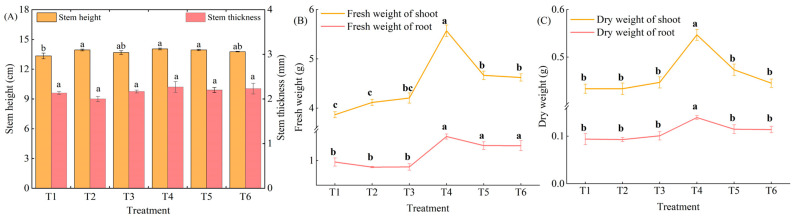
Effect of different concentrations of MeJA on stem height and stem thickness of pepper seedlings under LL stress. T1, 0 μM MeJA. T2, 100 μM MeJA. T3, 150 μM MeJA. T4, 200 μM MeJA. T5, 250 μM MeJA. T6, 300 μM MeJA. The results are expressed as the mean ± SE of five replicates, and the different letters denote the significant difference among treatments (*p* < 0.05), according to Duncan’s multiple tests. (**A**) Stem height and stem thickness. (**B**) Fresh weight. (**C**) Dry weight.

**Figure 2 plants-13-02694-f002:**
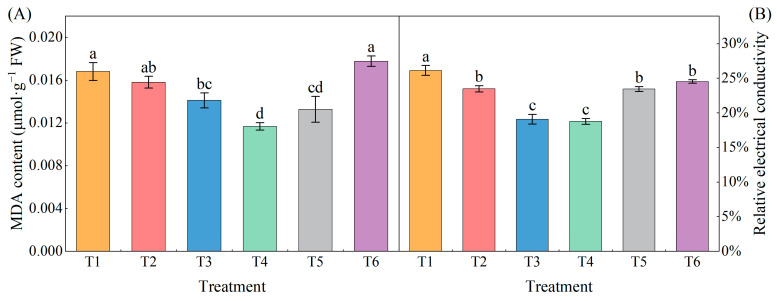
Effects of different concentrations of MeJA on MDA and relative conductivity of pepper seedlings under low temperature combined with low-light stress. The results are expressed as the mean ± SE of five replicates, and the different letters denote the significant difference among treatments (*p* < 0.05), according to Duncan’s multiple tests. (**A**) MDA content. (**B**) Relative electrical conductivity.

**Figure 3 plants-13-02694-f003:**
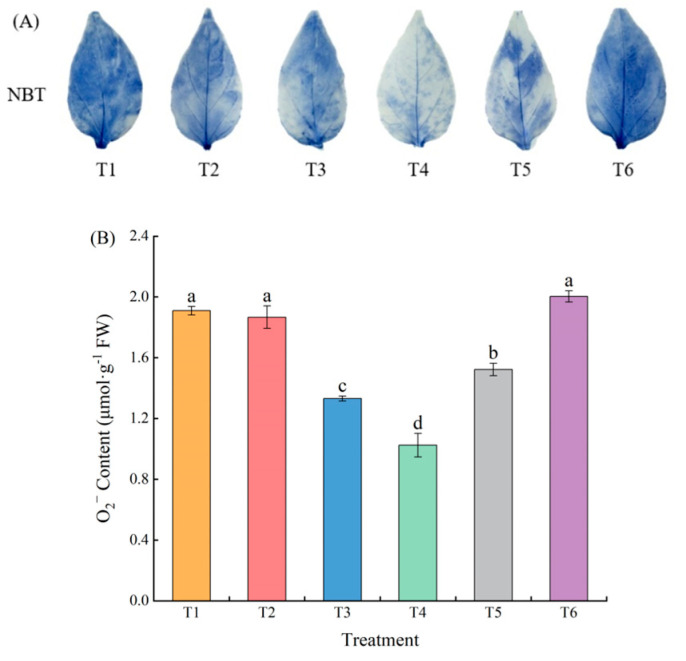
Effect of different concentrations of MeJA on the superoxide anion content of pepper seedlings under low temperature combined with low-light stress: The results are expressed as the mean ± SE of five replicates, and the different letters denote the significant difference among treatments (*p* < 0.05), according to Duncan’s multiple tests. (**A**) NBT histochemical staining. (**B**) O_2_^−^ content.

**Figure 4 plants-13-02694-f004:**
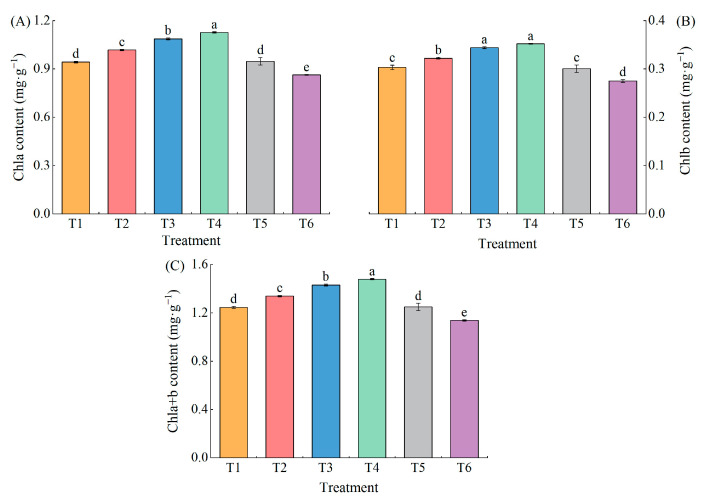
Effect of different concentrations of MeJA on chlorophyll content of pepper seedlings under low temperature combined with low-light stress. The results are expressed as the mean ± SE of five replicates, and the different letters denote the significant difference among treatments (*p* < 0.05), according to Duncan’s multiple tests. (**A**) Chlorophyll a content. (**B**) Chlorophyll b content. (**C**) Chlorophyll a+b content.

**Figure 5 plants-13-02694-f005:**
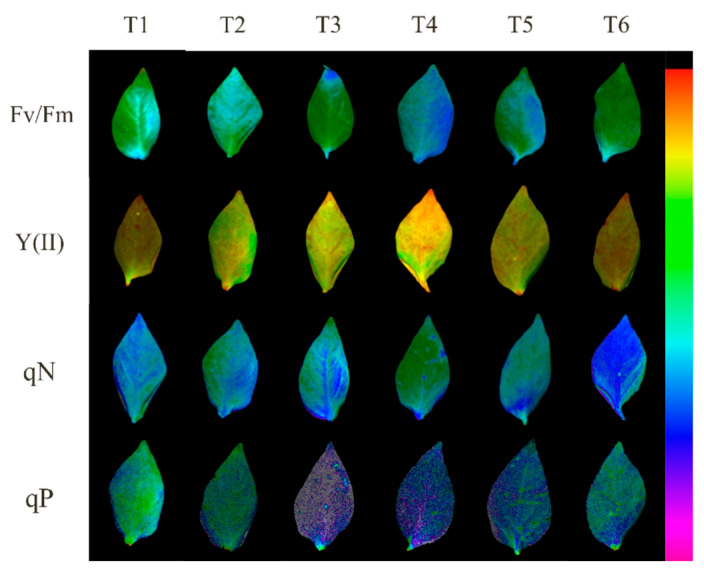
Visual analysis of chlorophyll fluorescence parameters of low temperature combined with low light pepper seedlings treated with different concentrations of MeJA. Images of Fv/Fm, Y(II), qP and qN.

**Figure 6 plants-13-02694-f006:**
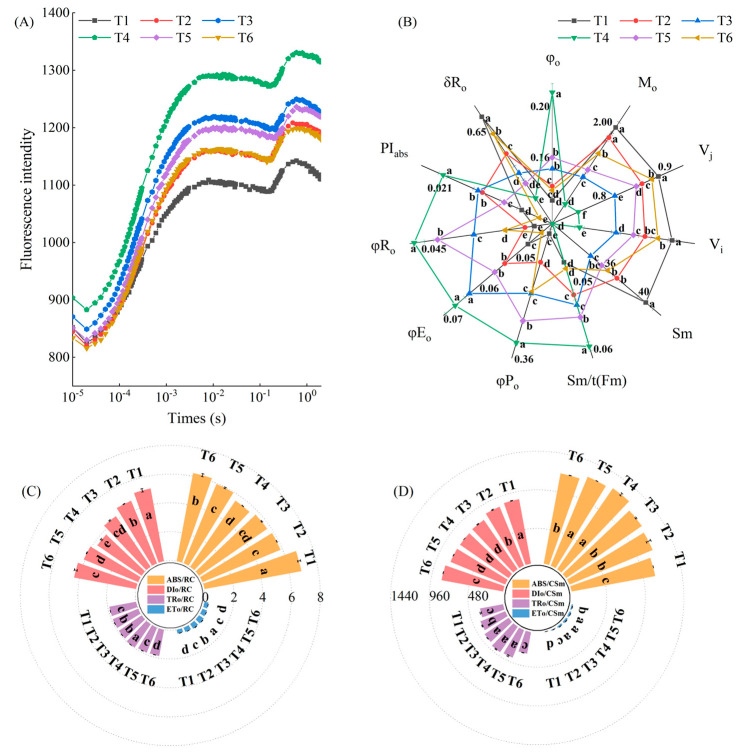
Effects of different concentrations of MeJA on chlorophyll fluorescence kinetics (OJIP curves), JIP-test parameters of pepper seedlings, energy allocation to individual active reaction centers, and energy allocation per unit cross section in pepper seedlings under low temperature combined with low-light stress: The results are expressed as the mean ± SE of five replicates, and the different letters denote the significant difference among treatments (*p* < 0.05), according to Duncan’s multiple tests. (**A**) OJIP curves. (**B**) JIP-test parameters. (**C**) Electron absorption, capture, transfer, and dissipation energy per unit active reaction center. (**D**) Electron absorption, capture, transfer, and dissipation energies per unit cross section.

**Figure 7 plants-13-02694-f007:**
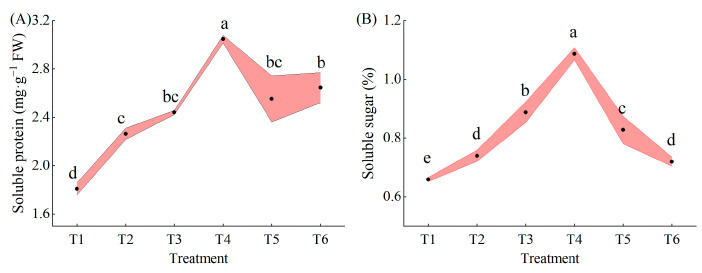
Effects of different concentrations of MeJA on soluble proteins and soluble sugars in pepper seedlings at low temperature combined with low-light stress. The results are expressed as the mean ± SE of five replicates, and the different letters denote the significant difference among treatments (*p* < 0.05), according to Duncan’s multiple tests. (**A**) Soluble protein content. (**B**) Soluble sugar content.

**Figure 8 plants-13-02694-f008:**
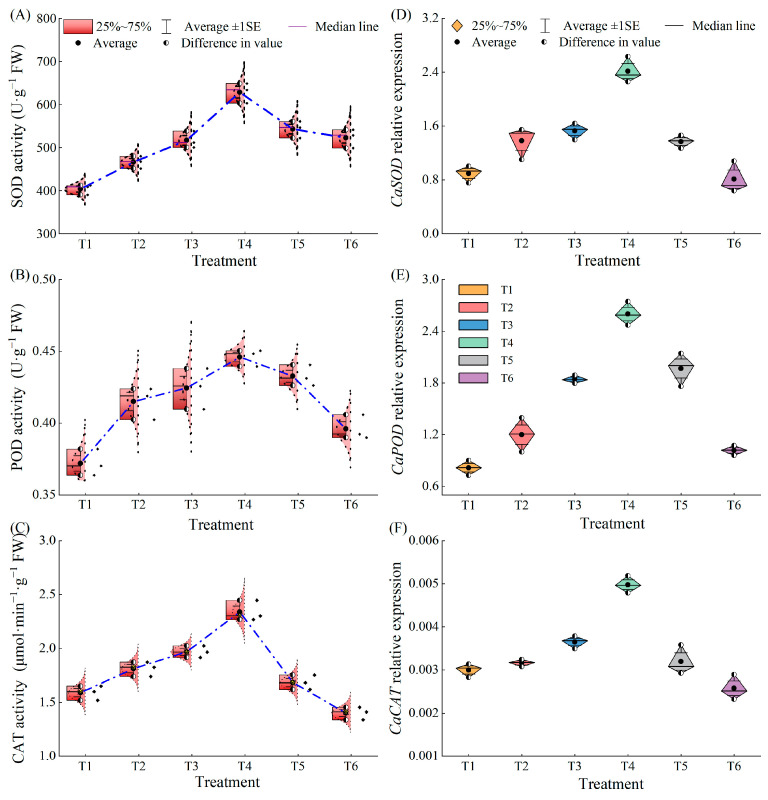
Effects of different concentrations of MeJA on antioxidant enzyme activities and expression of related genes in pepper seedlings under low temperature combined with low-light stress: (**A**–**C**) SOD, POD, CAT activity. (**D**–**F**) *CaSOD*, *CaPOD*, *CaCAT* relative expression.

**Table 1 plants-13-02694-t001:** Effects of different concentrations of MeJA on chlorophyll fluorescence parameters of peppers under low temperature combined with low-light stress.

Treatment	Fv/Fm	Y(II)	NPQ	qN	qP	1-qP	(1-qP)/NPQ
T1	0.374 ± 0.007 de	0.173 ± 0.006 c	0.259 ± 0.008 a	0.679 ± 0.010 b	0.493 ± 0.011 d	0.507 ± 0.011 a	1.964 ± 0.100 a
T2	0.412 ± 0.009 cd	0.191 ± 0.006 b	0.251 ± 0.005 a	0.570 ± 0.003 c	0.521 ± 0.022 d	0.479 ± 0.022 a	1.909 ± 0.077 a
T3	0.454 ± 0.007 bc	0.207 ± 0.002 a	0.185 ± 0.008 b	0.586 ± 0.016 c	0.673 ± 0.003 b	0.327 ± 0.003 c	1.774 ± 0.099 a
T4	0.586 ± 0.008 a	0.218 ± 0.003 a	0.070 ± 0.003 d	0.458 ± 0.007 e	0.937 ± 0.020 a	0.063 ± 0.020 d	0.928 ± 0.311 b
T5	0.472 ± 0.037 b	0.190 ± 0.005 b	0.147 ± 0.007 c	0.517 ± 0.004 d	0.706 ± 0.027 b	0.294 ± 0.027 c	2.017 ± 0.234 a
T6	0.341 ± 0.019 e	0.171 ± 0.003 c	0.186 ± 0.006 b	0.719 ± 0.007 a	0.575 ± 0.007 c	0.425 ± 0.007 b	2.287 ± 0.060 a

Note: T1, 0 μM MeJA. T2, 100 μM MeJA. T3, 150 μM MeJA. T4, 200 μM MeJA. T5, 250 μM MeJA. T6, 300 μM MeJA. The results are expressed as the mean ± SE of three replicates, and the different letters denote the significant difference among treatments (*p* < 0.05), according to Duncan’s multiple tests.

**Table 2 plants-13-02694-t002:** JIP-test indicates the meanings of related parameters.

Parameters	Significance
Fv/Fm	Maximum photochemical efficiency of PSII
PⅠ (abs)	Performance index based on absorbed light energy
Vj	J-point relative variable fluorescence
Vi	I-point relative variable fluorescence
Sm	Normalized total residual region above the O-J-I-P transient
φPo	Maximum quantum yield of primary PSII photochemistry
φEo	Quantum yield of electron transport flux from QA to QB
φRo	Quantum yield of PSII final electron acceptor reduction per photon absorbed
PI (abs)	Performance index based on absorbed light energy
ABS/RC	Average absorbed light quantum flux per PSII reaction center
TRo/RC	Maximum captured photonic flux per active RC
DIo/RC	Light quantum flux dissipated per active RC
ETo/RC	Light quantum flux per active RC electron transfer
ABS/CSm	Energy flux absorbed in the excitation cross section (CSm)
TRo/CSm	Energy flux captured by absorption in the excitation cross section (CSm)
DIo/CSm	Energy flux dissipated in the excitation cross section (CSm)
ETo/CSm	Energy flux for electron transfer in the excitation cross section (CSm)

**Table 3 plants-13-02694-t003:** The sequences of primers used for the qRT-PCR.

Gene Name	Sequence (5′-3′)	GenBank Accession Number	Amplicons Size (bp)
*CaSOD*	F: GTGAGCCTCCAAAGGGTTCTCTTG	AF036936.2:35-721	127
	R: AAACCAAGCCACACCCAACCAG		
*CaPOD*	F: GCCAGGACAGCAAGCCAAGG	FJ596178.1:1:68-1042	131
	R: TGAGCACCTGATAAGGCAACCATG		
*CaCAT*	F: TTAACGCTCCCAAGTGTGCTCATC	NM_001324674.1:72-1550	116
	R: GGCAGGACGACAAGGATCAAACC		
*Actin*	F: GTCCTTCCATCGTCCACAGG	XM_016722297.1	133
	R: GAAGGGCAAAGGTTCACAACA		

Note: F: forward primer; R: reverse primer.

## Data Availability

Data are contained within the article.

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
