# Peer review of "Alleviating Effects of Methyl Jasmonate on Pepper (Capsicum annuum L.) Seedlings under Low-Temperature Combined with Low-Light Stress"

_plants, 2024, doi:10.3390/plants13192694_

Round 1
Reviewer 1 Report
Comments and Suggestions for Authors
The manuscript concerns the activity of methyl jasmonate in alleviating low temperature and low light stress in pepper. The article is important in the field of pepper cultivation, but it has some flaws that need to be corrected. The details are listed below:
L37: Latin names in italics. Correct throughout the paper
L45: deteriorate instead affect
L85: stem height and stem thickness in the legend to Fig. 1A
L213: it seems that down-regulated CaSOD is only in T6 compared to T1
L214: it seems that only CaCAT did not change significantly
L329: emphasize the ubiquitous role of antioxidant enzymes in mitigating different abiotic stress (pesticides, heavy metals, drought, low temperature and light) and biotic stress (pests, phytopathogenic fungi). For this purpose refer to: https://doi.org/10.1016/j.chemosphere.2022.136284 and https://doi.org/10.3390/agronomy13051378
L385-388: in what growth stage the seedlings were sprayed with MeJA and harvested
L388: how plants were sprayed, with hand sprayer? What was the volume of MeJA solution per treatment? How many pots per treatment were used?
L405: change the heading to ‘Physiological and biochemical parameters’ or similar
L444: why OD at 470 nm was recorded? This wavelength is specific for carotenoids, but these pigments were not included in the study
Reviewer 2 Report
Comments and Suggestions for Authors
Plants 3206903
Comment to authors
The authors carried out an in-depth analysis of the effects of methyl jasmonate (MeJA) on the photosynthetic apparatus and the activity of enzymes involved in the defence against abiotic stresses in pepper seedlings under low light and temperature. The methodological section lacks a description of the experimental setup and measurement conditions. In the discussion section, some parts are difficult to understand and need to be checked and corrected. The form of the references is not correct in the text or in the list.
Detailed comments
The full name of the methyl jasmonate should also be given in the title, abstract and in the Materials and Methods section. In addition, the full names of the abbreviations in lines 18-19 of the abstract should be printed.
Materials and methods section needs improvement.
In each subsection, the number of samples of plants or leaves and their condition (fresh or frozen) should be clearly described:
-How many seedlings were in each treatment and used for the measurements.
-Lines 400-404: after 7 days of treatment (low temperature and light) was the whole plant frozen or just the leaves? What tests were performed on the frozen material?
-To determine the dry matter content, the fresh matter is required (line 406), in lines 410-412 the measurements also refer to fresh material.
-How many plants were used for chemical analysis? see lines 410, 429, 440
Results:
-The comment under the figures should be amended and the separate comment on the sub-figures should be deleted.
e.g. under fig 1: "effect of different MeJA concentrations on plant height and stem thickness (A), fresh weight (B) and dry weight (C) of pepper seedlings under LL stress", then delete the comment in line 90. Similar changes should be made to Figures 2,4, and 7
-Fig. 4C: It seems to me that the total chlorophyll (a+b) content is the same for T1 and T5 treatments but lower for T6 than for T1. Check the marking of significant differences in Figure 4c.
-Line 209: check the sentence, I think you should delete "compared with the T1 treatment" after the brackets.
-Lines 213-214- The sentence is not understandable. What did you want to say?
-Lines 337-342: This part is difficult to understand. Check and revise.
-Lines 357- 364: Sentence is too long and difficult to follow.
-In the text of citations, the reference number should be given in parentheses after the name in the following format e.g. Lie et al. (18), Yu et al. (24)… Correct throughout the manuscript.
Others:
Line 328: what does TBARS stand for? to be written out
Line 420: TCA?
Line 429: NBT?
Line 432: PBS?
Line 444: OD?
-Reference list should be completely improved. For articles with multiple authors, the et al. notation is not enough. The names of co-authors should also be given.
Comments on the Quality of English LanguageSome sentences are difficult to understand so it is necessary to check and correct.
Round 2
Reviewer 1 Report
Comments and Suggestions for Authors
The Authors have improved the paper. I have no more comments.
Reviewer 2 Report
Comments and Suggestions for Authors
Plants-3206903 revised
Comments to authors
The authors have improved the manuscript, taking into account the suggestions, but the correction of the reference list has not been done well. Probably my suggestion on this was not clear enough.
When citing articles with multiple authors in the text, it is sufficient to give the name of the first author and the et al (year) which has been correctly corrected in the manuscript. In the reference list, however, the name of each author should be written out and corrected.
Correct the reference list everywhere as follows:
For multiple authors:
e.g. 1. Bohra, A., Gahlaut, V., Perovic, D., Varshney, RK. Genetics and epigenetics: Plausible role in development of climate resilient crops. 2023, Frontiers Media SA. p. 561 1165843.
3. Hasanuzzaman, M., Al Mahmud, J., Anee, TI., Nahar, K., Islam. T. Drought stress tolerance in wheat: omics approaches in understanding and enhancing antioxidant defense. In book: Abiotic stress-mediated sensing and signaling in plants: an omics perspective, 2018: p. 267-307.
But the two-author articles should be as follows:
2. Boguszewska, D., B. ZagdaÅ„ska. ROS as signaling molecules and enzymes of plant response to unfavorable environmental conditions. Oxidative stress–molecular mechanisms and biological effects. Rijeka, Croatia: InTech, 2012: p. 341-362.
9. Zaid, A., F. Mohammad, Methyl jasmonate and nitrogen interact to alleviate cadmium stress in Mentha arvensis by regulating physio-biochemical damages and ROS detoxification. Journal of Plant Growth Regulation, 2018. 37(4): p. 1331-1348.
